# Exploiting the Stemness and Chemoresistance Transcriptome of Ewing Sarcoma to Identify Candidate Therapeutic Targets and Drug-Repurposing Candidates

**DOI:** 10.3390/cancers15030769

**Published:** 2023-01-26

**Authors:** Elizabeth Ann Roundhill, Pan Pantziarka, Danielle E. Liddle, Lucy A. Shaw, Ghadeer Albadrani, Susan Ann Burchill

**Affiliations:** 1Children’s Cancer Research Group, Leeds Institute of Medical Research, St James’s University Hospital, Beckett Street, Leeds LS9 7TF, UK; 2Anticancer Fund, Brusselsesteenweg 11, 1860 Meise, Belgium

**Keywords:** Ewing sarcoma, stemness, pluripotency, CD133, multidrug resistance, ABCG1, POU5F1/CT-4, drug repurposing

## Abstract

**Simple Summary:**

Ewing sarcoma is a cancer arising most frequently in teenagers and young adults. For many patients, outcomes are the same today as they were 30 years ago, emphasising the need for more effective treatments to eradicate the cells responsible for progression and relapse. These cells responsible for progression and relapse have been identified using assays that evaluate functional characteristics and expression of cell surface markers. For the first time, we reveal ABCG1 as an additional potential cell surface marker of progression. In rare cancers like Ewing sarcoma, commercial development of new drugs is seldom a priority, reflecting the small number of patients and lack of well-characterised molecular subtypes. Therefore, rather than creating new drugs, which can take 20 to 30 years, repurposing of existing drugs may be an efficient cost-effective strategy to accelerate novel molecularly targeted therapy into clinical trials for patients with Ewing sarcoma. To identify candidate molecular targets, we have used a combination of functional assays and transcriptomic analyses to characterize the cells responsible for progression and relapse. We have then applied a bespoke in silico pipeline to find drugs with known safety profiles that bind to these targets. In the future, after preclinical validation of efficacy and specificity in Ewing sarcoma, some of these drugs may be assessed as combination treatments in clinical trials, with the goal of improving outcomes.

**Abstract:**

Outcomes for most patients with Ewing sarcoma (ES) have remained unchanged for the last 30 years, emphasising the need for more effective and tolerable treatments. We have hypothesised that using small-molecule inhibitors to kill the self-renewing chemotherapy-resistant cells (Ewing sarcoma cancer stem-like cells; ES-CSCs) responsible for progression and relapse could improve outcomes and minimise treatment-induced morbidities. For the first time, we demonstrate that ABCG1, a potential oncogene in some cancers, is highly expressed in ES-CSCs independently of CD133. Using functional models, transcriptomics and a bespoke in silico drug-repurposing pipeline, we have prioritised a group of tractable small-molecule inhibitors for further preclinical studies. Consistent with the cellular origin of ES, 21 candidate molecular targets of pluripotency, stemness and chemoresistance were identified. Small-molecule inhibitors to 13 of the 21 molecular targets (62%) were identified. POU5F1/OCT4 was the most promising new therapeutic target in Ewing sarcoma, interacting with 10 of the 21 prioritised molecular targets and meriting further study. The majority of small-molecule inhibitors (72%) target one of two drug efflux proteins, p-glycoprotein (*n* = 168) or MRP1 (*n* = 13). In summary, we have identified a novel cell surface marker of ES-CSCs and cancer/non-cancer drugs to targets expressed by these cells that are worthy of further preclinical evaluation. If effective in preclinical models, these drugs and drug combinations might be repurposed for clinical evaluation in patients with ES.

## 1. Introduction

Ewing sarcoma (ES) arises in the bone or soft tissue [1], most frequently presenting in young people aged 10–25 years [2]. Standard of care treatment, including a combination of multi-agent chemotherapy, surgery and radiotherapy, has improved outcomes for some patients [1]. However, around 40% of these patients develop multidrug-resistant (MDR) metastatic disease [2,3,4,5], leading to relapse and survival rates normally associated with metastatic disease (5-year survival 10% [5,6,7]). Late relapses and chemotherapy-induced morbidities are additional enduring burdens for patients, their families and carers. A major unmet clinical need is therefore the introduction of molecularly targeted therapy to minimise treatment-related toxicity and improve outcomes.

Progression and relapse are driven by subpopulations of cells capable of self-renewal and migration within tumours that are resistant to current treatments. These cancer stem-like cells (CSCs) have been identified in a range of adult and paediatric solid tumours based on expression of cell surface markers, most frequently CD133 (also known as prominim-1) [8,9,10,11,12]. Ewing sarcoma cancer stem-like cells (ES-CSCs) have been isolated based on expression of CD133 [13,14,15]. However, distinct CD133-negative CSCs are present in some cancers [8,9,10,11,12], including Ewing sarcoma [15]. Therefore, additional approaches, including formation of clones from single cells and generation of three-dimensional (3D) spheroids, have been used to improve the identification of ES-CSC [16,17] and CSC in other cancer types [18,19,20].

In this study, we have investigated the completeness of CD133 as a cell surface marker of ES-CSCs and identified ABCG1 as an additional marker of these cells using 3D spheroids and self-renewal from single cells. The expression and prognostic potential of this ABC transporter protein has been evaluated in patient-derived cells and tumours in the online dataset GSE17618. Genes that regulate pluripotency, stemness and the MDR ABC transporter proteins have been identified by comparing the transcriptomes of substrate adherent two-dimensional (2D) non-CSC and 3D spheroid-derived ES-CSCs. We combined genes that were differentially expressed in ES cells grown in 3D spheroids and cells grown in 2D with genes previously reported in patient-derived ES-CSCs [17] to pinpoint candidate molecular targets expressed by ES-CSCs. We then developed a bespoke pipeline to identify small-molecule inhibitors of these targets for further preclinical studies. If these drugs are effective in preclinical models of ES, they may in the future be accelerated into clinical trials for evaluation in patients. This repurposing strategy seeks to reuse existing licensed drugs to treat new indications and is a complementary strategy to de novo drug development [21,22].

## 2. Materials and Methods

### 2.1. Cell Culture

ES cell lines (A673, RD-ES, SKES-1, SK-N-MC, TC-32 and TTC-466) were cultured as previously described [23] and purchased from the American Type Culture Collection, Manassas, VA, except for the following cells that were kind gifts: TC-32 cells from Dr. J. Toretsky (Division of Pediatrics, University of Maryland, Baltimore, MD, USA), the TTC-466 cells from Dr. P. Sorenson (British Columbia Children’s Hospital, Vancouver, BC, Canada). Primary ES cell cultures and daughter ES-CSCs were cultured as previously described [17]. The embryonic stem cell (ESC) culture SHEF-4 (RRID:CVCL_9791) was a gift from Professor P. Andrews, Centre for Stem Cell Biology, University of Sheffield, Sheffield, UK [24] and used as a positive control for CD133 and ATP binding cassette subfamily G member 1 (ABCG1). The glioblastoma cell line (T98G) was a gift from Professor M. Knowles, University of Leeds, and was a positive control for multidrug resistance-associated protein 1 (MRP1, gene name = ABCC1) [23]. The human embryonic fibroblast line KMST-6 (cultured in MEM containing 10% FBS and 2 mM glutamine) and Jurkat cells (cultured in RPMI 1640 containing 10% FBS and 2 mM glutamine) were gifts from Dr E. Morrison, University of Leeds, and used as the positive control and to generate the calibration curve, respectively, for the measurement of telomere length.

### 2.2. Western Blotting

Western blotting (WB) was performed as previously described [23]. Equal loading of proteins was confirmed using α-tubulin [23] or β-actin (0.4 µg/mL, A5441, Sigma-Aldrich, Paisley, UK). For the detection of CD133, protein extracts were heated at 95 °C for 5 min before cooling on ice and WB (CD133, 1 µg/mL, W6B3C1, Miltenyi Biotech, Surrey, UK). After incubation with primary antibodies (MRP1 [23] or ABCG1 (1:1000, ab36969; Abcam Plc., Cambridge, UK)) and secondary antibodies [23], proteins were visualised by image capture (at different exposure times depending on the intensity of signal) and quantified using the Li-cor Odyssey infrared imaging system (Li-cor Biosciences, Lincoln, NE, USA).

### 2.3. Flow Cytometry

#### 2.3.1. Cell Surface Expression of CD133

Cells (5 × 10^5^) were incubated in FcR blocking buffer (Miltenyi Biotech) and either anti-CD133/2 (4.5 µg/mL, clone 293C3, Miltenyi Biotech) or the isotype control IgG2b-PE (4.5 µg/mL, Miltenyi Biotech) antibodies in the dark at 4 °C for 10 min. Cells (10,000 per sample) were then analysed by flow cytometry using the FACSCalibur (BD Biosciences, Berkshire, UK).

#### 2.3.2. Co-Expression of ABCG1 and CD133

SK-N-MC cells were incubated in normal goat serum (1:10, Dako, Agilent Technologies, Santa Clara, CA, USA) in BD Perm/Wash™ Buffer for 30 min at 4 °C. Cells were then incubated with CD133 (anti-CD133/2, 4.5 µg/mL, clone 293C3) and/or ABCG1 rabbit polyclonal antibody (100 µg/mL, PA5-13462, Thermo-Fisher Scientific, Paisley, UK) in BD Perm/Wash™ Buffer for 1h at 4 °C in the dark. Control cells were incubated with IgG2b-PE (4.5 µg/mL, Miltenyi Biotech) or rabbit IgG isotype control antibody (100 µg/mL, Dako). Cells were then incubated with the secondary antibody (2 μg/mL goat anti-rabbit IgG FITC, A31556, Thermo-Fisher Scientific) in BD Perm/Wash™ Buffer for 30 min at 4 °C in the dark. Cells (10,000 per sample) were analysed by flow cytometry using the CytoFLEX (Beckman Coulter, High Wycombe, UK).

### 2.4. Self-Renewing Ability

Growth of progeny from a single cell as an adherent culture was determined as previously described [17,18]. A single cell (Poisson distribution probability of λ < 1 = 0.9) was seeded into each well of 10 Primaria^TM^ 96-well plates (Corning) and the number of wells containing ≥5 cells was recorded after 21 days by light microscopy (Olympus CKX41). Where possible, single cell self-renewing cell populations were propagated to establish daughter cell cultures; these are subsequently referred to as ES-CSCs. To examine colony forming efficiency in soft agar, a single cell suspension (1.8 × 10^5^ cells) in cell-specific media containing 0.3% agar (Aldrich, Poole, UK), was overlaid on a solid agar bed (0.5% agar in media). After 21 days, colonies were stained with 8 mM iodonitrotetrazolium chloride (made up in ddH_2_O (*w*/*v*); Sigma-Aldrich) for 16 h and colony number counted by light microscopy. Colony forming efficiency = [the number of colonies formed in the field of view/number of cells seeded] × 100.

For spheroid formation, a single cell was seeded into each well of an ultra-low attachment plate (Corning, UK) in cell line-specific media. The number of spheroids at 21 days was recorded to calculate the spheroid forming efficiency (SFE); SFE = [number of wells containing a spheroid/the number of wells seeded with a single cell] × 100. Spheroids were imaged by light microscopy (Olympus CKX41). For reverse transcriptase quantitative polymerase chain reaction (RTqPCR), Western blot, immunocytochemistry (ICC) and flow cytometry, spheroids formed from single cells were collected at 21 days.

### 2.5. Magnetically Activated Cell Sorting for the Enrichment of CD133-Positive Cells

ES cells (1 × 10^8^) were incubated with 300 µL of CD133 microbeads and 100 µL of FcR blocking buffer (Miltenyi Biotech). CD133-positive cells were isolated using LS columns (Miltenyi Biotech). CD133-negative cells were depleted of labelled (CD133-positive) cells by passing the cells through two LD columns (Miltenyi Biotech). Cell surface expression of CD133 was confirmed by flow cytometry immediately after separation: >90% of the CD133-positive selection expressed CD133, and CD133 was expressed by <5% of the CD133-negative selected cells. CD133-positive A673 and TC-32 cells remained positive over 15 passages.

### 2.6. RNA Expression of Markers of Pluripotency and Differentiation, the Wnt Signalling Pathway and ABC Transporter Proteins

RNA was extracted using the RNeasy Micro Kit (Qiagen, Düsseldorf, Germany) and RNA quantity and quality measured using the Nanodrop and Agilent 2100 bioanalyzer. RNA (with a RIN > 8) was converted to cDNA by reverse transcription [23]. The mRNA expression was evaluated using the TaqMan^®^ Human Stem Cell Pluripotency Array, TaqMan^®^ Human Wnt Pathway and TaqMan^®^ Human ABC Transporter Array (Applied Biosystems, Waltham, MA, USA) [23]. To allow direct comparison across the 3 array platforms target Ct values were normalised using the global mean [25,26,27]. ABC transporter and pluripotency mRNAs (*n* = 140, excluding endogenous control mRNAs) more highly expressed than other mRNAs (Ct values < 25), expressed (Ct values 25–35) and not expressed (Ct values >35, [18,28,29]) in 3D spheroids from TTC 466 and SK-N-MC cells were identified. Unique and shared mRNAs in each group were analysed using the Search Tool for Retrieval of Interacting Genes/Proteins (STRING) database (http://string-db.org, [17,30]) to identify GO terms. Significant differences in mRNA expression were determined using Linear Models for Microarray Data (LIMMA) [18]. Target mRNAs were validated using individual RTqPCR assays if the Q value was <0.1, there was at least a mean Ct difference of >2 between the compared populations and Ct values were <35 [18]. For mRNA validation, total RNA (20 ng) was reverse-transcribed and cDNA added to the PCR mix containing sequence specific reverse and forward primers and probe for PPIA (the housekeeping gene) or ABCG1 (Thermo Fisher Scientific; ABCG1 Hs00245154_m1) and 1 × TaqMan^®^ Universal PCR Master Mix (Thermo Fisher Scientific). RNA expression was calculated using the comparative Ct method [31].

ABCG1 transcripts were characterised in total RNA extracted from SK-N-MC cells grown as 2D cultures or 3D spheroids, sequenced and aligned as previously described [17]. Normalised read counts were identified in total RNA sequencing data using DESeq2 [32], adjusted *p* values of <0.01 were considered significant.

### 2.7. Immunohistochemistry (IHC)

Formalin-fixed paraffin-embedded (FFPE) sections (4 µm) of spheroids were deparaffinised in xylene (2 min) and rehydrated before antigen retrieval in citric acid buffer (10 mM in ddH_2_O, pH6, [33]). Endogenous peroxidases were blocked using 3% hydrogen peroxide for 5 min and endogenous biotin, biotin receptors and avidin-binding sites blocked using an avidin/biotin blocking kit (Invitrogen, Waltham, MA, USA). Sections were incubated for 1 h with ABCG1 primary antibody (10 µg/mL, PA5-13462, Thermo-Fisher Scientific) or rabbit IgG control (10 µg/mL, Dako) at room temperature, followed by incubation with the secondary antibody. Sections were then incubated with streptavidin–peroxidase (Abcam Plc.), followed by DAB substrate (Dako) for 10 min and nuclei counterstained with haematoxylin.

### 2.8. siRNA Knockdown of ABCG1

Cells (5 × 10^4^) were seeded and allowed to adhere overnight. Media were replaced with Accell siRNA Delivery Media (B-005000-500, Dharmacon, Lafayette, CO, USA) alone or containing ABCG1 siRNA (1.5 µM, E-008615-00-0005, SMARTpool:Accell ABCG1 siRNA, containing 4 siRNAs targeting exon 6, present in all canonical and novel transcripts (Dharmacon) or non-targeting control siRNAs (1.5 µM, D-001910-10-20, Dharmacon) for 72 h. ABCG1 knockdown was confirmed by RTqPCR.

### 2.9. Apoptosis

Cells (5 × 10^4^) were harvested following treatment with ABCG1 and control siRNA for 24–72 h and apoptosis measured by annexin V and propidium iodide (PI) labelling of cells (annexin V–FITC apoptosis detection kit, BD Biosciences, [34]).

### 2.10. Statistical Analyses

Differences in proliferation and viable cell numbers were log transformed and analysed by linear regression. Differences in the gradients of each plot were compared using the extra sum of squares F test. For all other experiments, data were analysed by analysis of variance (ANOVA followed by a Bonferroni’s or Tukey’s post hoc test), or a non-parametric Mann–Whitney or unpaired two-tailed *t*-test. Correlations were determined using a Pearson’s correlation coefficient (r). *p* values of <0.05 were considered significantly different. Statistical analyses were performed using GraphPad PRISM 7.03 (GraphPad Software, San Diego, CA, USA).

### 2.11. Identification of Candidate Drugs for Prioritised Molecular Targets

Gene lists were analysed using the STRING database. To identify drugs reported to target these molecular candidates, we interrogated DrugBank (version 5.1.9 [35]) and DGI database (DGIdb, version v4.2.0 [36]) to access information on FDA-approved drugs and their molecular targets. Additional drug-target information was derived from the literature-based repurposing drugs in oncology (ReDO) database [37]. A polypharmacology approach identified targets for each drug candidate. Those targets previously associated with ES using the Open Targets Platform [38] and DisGeNET [39] were employed as the identification source. Data from DGIdb and Open Targets characterise the strength of target–disease and drug–target associations. Using DGIdb, the interaction score is computed from: (publication count + data source count) × (average known gene partners for all drugs/known gene partners for candidate drug) × (average known drug partners for all genes/known drug partners for target gene). The Dir DGI score therefore incorporates data on the number of molecular targets of the drug, the number of drugs that target a gene and the number of publications and data sources supporting the association. The Dir Assoc score is generated from the Open Targets database and is a measure of the relationship between the molecular target and “cancer” as the disease term.

Licensed cancer drugs were identified from the list of drug candidates addressing >1 ES molecular target using the Cancer Drugs database [40]. ES trials were identified using clinicaltrials.gov, EU Clinical Trials Register and WHO International Clinical Trials Registry Platform. For non-cancer drugs previously identified as oncological repurposing candidates (ReDO database), clinical trial activity in any cancer was extracted using the ReDO_Trials database of active repurposing trials in oncology [41]. For these repurposing candidates, a support score was calculated from data in the ReDO database to characterise the range of data available illustrating the anticancer effects of the drugs (e.g., in vitro, in vivo, case reports, observational data and clinical trial data).

## 3. Results

### 3.1. ES Cell Lines Produce Spheroids and Clones from a Single Cell

All ES cell lines (6/6) formed clones or spheroids from a single cell when cultured in soft agar or on ultra-low attachment plates (Table 1, Figure 1), consistent with previous reports that ES cell cultures contain a self-renewing cell population [8,9,10,11,12,13,14,17].

The formation of single cell-derived clones and spheroids in soft agar (29 ± 2%, *p* < 0.05) and ultra-low attachment conditions (75 ± 7%, *p* < 0.05; Table 1, Figure 1) was most efficient in the SKES-1 cell line. Spheroid formation was 100% in all cell lines when combining 100 cells or more (Table 1). Larger spheroids (>400 µm), produced cell line-specific morphology (Figure 1). SKES-1 spheroids produced a relatively uniform sphere of disseminated or dissociated cells, consistent with spheroid formation in a range of solid tumour cell types [42,43,44,45,46]. For the first time, we identified 3D projections developing from the central core in five of six ES cell lines (Figure 1). The biological significance of these projections requires further investigation.

### 3.2. CD133 Identifies Some Self-Renewing Drug-Resistant ES-CSCs

CD133 protein was detected in all ES cell lines, with the exception of SK-N-MC cells (Figure 2A and Appendix A). A673 and TC-32 CD133-positive cells formed significantly more colonies in soft agar than the CD133-negative SK-N-MC cells (Appendix A). However, there was no difference in proliferation, cell cycle status or telomere length across the two cell lines (Appendix A), phenotypes frequently associated with CSCs and self-renewing ability [47]. Moreover, expression and activity of the multidrug-resistant protein MRP1 was increased in CD133-positive TC-32 cells, but not the A673 CD133-positive population (Appendix A). The shared CSC phenotype of both CD133-positive and CD133-negative ES populations demonstrates that CD133 expression alone is not sufficient to enrich for all the ES-CSC population.

Consistently with the cellular origin of ES and the high level of stemness markers expressed by these tumour cells [48,49,50], no significant differentially expressed genes associated with pluripotency, ABC transporter and Wnt signalling pathways were identified in CD133-positive and CD133-negative cells (Figure 2B (TC-32) and Figure 2C (A673), Appendix A). These observations are also consistent with the premise that CD133 can be used to identify some cells with characteristics of ES-CSCs, although pathways classically associated with the CSC phenotype in other cancer types are also expressed in CD133-negative ES cells. To investigate the CD133-independent ES-CSC phenotype, we went on to investigate ES-CSCs enriched through spheroid formation in ES cells with no or low CD133 expression.

### 3.3. Gene Expression Profile of CD133 Low or Negative ES 3D Spheroids and 2D Cultures

As TTC 466 and SK-N-MC cells had low or no detectable CD133 protein (Figure 2A), we compared the transcriptome of TTC 466 and SK-N-MC 3D spheroids and 2D cultures (Figure 3A). Consistently with the premise that spheroid formation is a feature of CSCs, expression of the stemness markers LEFTB and LIN28 were significantly increased in SK-N-MC spheroids compared to cells in 2D culture, whereas expression of LAMA1, COL2A1, ACTC, GCG, SEMA3A and PODXL were significantly decreased (Q value < 0.1, Figure 3B,D, Appendix A). In TTC 466 spheroids, no markers were significantly increased; however, expression of FLT1 and GATA6 was significantly decreased compared to cells in 2D culture (Figure 3C,E, Appendix A).

Expression of 20 ABC transporter genes was significantly different in SK-N-MC spheroids compared to the cells in 2D cultures, although only ABCG1 and CFTR were differentially expressed greater than twofold (Figure 4A–C). However, ABC transporter genes, including ABCG1, were not significantly differentially expressed in TTC 466 cells grown in 2D or as 3D spheroids (Figure 4D–F). Comparison of the mRNAs in SK-N-MC and TTC 466 spheroids revealed that 35% of genes were undetected in both populations and an additional seven (5%) and six (4%) unique mRNAs were undetected in TTC 466 and SK-N-MC spheroids respectively (Figure 4G). Fifty-six percent of mRNAs were detected in both spheroid populations and 8% of these were highly expressed (Figure 4G). The six mRNAs uniquely expressed in TTC 466 spheroids are involved in cell differentiation (GO:0030154), which may explain why increased expression of pluripotency mRNAs was not observed in TTC 466 spheroids. Previous studies have shown knockdown of *EWSR1-FLI1* decreases the expression of stemness markers and stem-like properties of ES cells [51]. Whether the lack of shared pluripotency mRNAs increased in the TTC 466 and SK-N-MC spheroids reflects the different transcriptional activators of TTC 466 (*EWSR1-ERG)* and SK-N-MC cells (*EWSR1-FLI1*) remains to be seen.

The decrease in expression of the CFTR gene was not validated using RTqPCR or Western blot (results not shown). However, the increase in ABCG1 expression was confirmed in SK-N-MC spheroids at the mRNA (*p* = 0.0015, Figure 4H) and protein (Figure 4I) level. ABCG1 is heterogeneously expressed in all ES cell lines (6/6; Appendix A). By IHC, ABCG1 expression was confirmed in the outer proliferating region of 3D SK-N-MC spheroids but not the hypoxic region or inner necrotic core (Figure 4J; [34]), suggesting expression of ABCG1 may be regulated by hypoxia and have a functional role in ES cells under these conditions.

### 3.4. Functional Role and Characterisation of ABCG1 in ES-CSCs

Knockdown of ABCG1 using siRNA had no effect on the viability or apoptosis of SK-N-MC cells in culture (10 ± 0.8% in non-targeting and 10 ± 1.1% in ABCG1 siRNA-treated cells, *p* > 0.05; Figure 5A). There was also no effect on proliferation (0.93 ± 0.1 and 0.94 ± 0.05 in non-targeting and ABCG1 siRNA treated cells respectively, *p* > 0.05) or self-renewing ability from a single cell in soft agar (60% in non-targeting and 58% in ABCG1 siRNA-treated cells). However, after knockdown of ABCG1 spheroid production was significantly reduced; 65 ± 3% in non-targeting and 33 ± 5% in ABCG1 siRNA-treated cells (*p* < 0.0001, *n* = 10). These data suggest that although ABCG1 may not have a role in homeostasis of ES cells in 2D, it may affect cell–cell interactions and components of the tumour microenvironment important in the development of 3D spheroids and possibly tumours. This hypothesis requires further investigation.

Analysis of ABCG1 RNA in SK-N-MC spheroids and 2D cultures using total RNA sequencing, revealed expression of 11 previously reported transcripts (www.ensembl.org, accessed on 27 April 2016; Figure 5B,C) and two novel ABCG1 RNA species (Figure 5C). Canonical ABCG1 generates the multiple transcripts by alternative splicing (Figure 5C). Novel transcript 1 is most similar to transcript 4 (ENST00000450121.5), both sequences missing exon 5 (Figure 5C). However, novel transcript 1 included exons 8–15 and had an extended 3′UTR region predicted to produce a larger molecular weight protein than transcript 4. The RNA sequence of novel transcript 2 is most similar to transcript 5 (ENST00000361802.6), the additional 3′UTR sequence unlikely to produce a unique protein product. Expression of both novel transcripts at the RNA level was increased above canonical protein producing sequences (transcripts 1, 3–8) in SK-N-MC cells grown as spheroids compared to 2D cultures, this was most significant for novel transcript 1 (Figure 5D). The increase in ABCG1 gene expression in SK-N-MC cells from spheroids (Figure 4G,H), might then reflect expression of novel transcript 1. Further studies are required to investigate this novel ABCG1 transcript and its role in ES-CSCs.

Since ABCG1 mRNA was increased in ES-CSCs from SK-N-MC spheroids, we examined its co-expression with CD133. There was no significant difference in the percentage of cells expressing ABCG1 or CD133 in SK-N-MC cells from 3D spheroids or 2D cultures (Figure 6A,B). Consistently with the increase in ABCG1 in protein extracts from 3D spheroids (Figure 4H), the level of ABCG1 protein per cell was greater in cells from 3D spheroids compared to 2D cultures (2.5 ± 0.8-fold increase, *p* < 0.05, Figure 6C). Levels of CD133 per cell were also increased (6.3 ± 1 fold increase, *p* < 0.05, Figure 6C), consistent with the higher expression of ABCG1 protein in TC-32 CD133-positive cells compared to CD133-negative cells (Figure 6D). Expression of ABCG1 RNA was confirmed at the protein level in primary patient-derived ES cells (Figure 6E). However, there was no correlation between the percentage of CD133-positive cell lines or patient-derived cells and progeny-producing ability measured using the colony formation assay in soft agar (R^2^ < 0.1) or self-renewing ability from a single cell seeded on low adherent or adherent plates (R^2^ = 0.002; Figure 6F), suggesting in patient-derived cells CD133 may not identify cell populations with a self-renewing phenotype.

Interrogation of the publicly available GSE17618 RNA dataset revealed high expression of CD133, but not ABCG1, was associated with a more than threefold risk of an event and poor outcome (Appendix A), consistent with the hypothesis that high expression of CD133 identifies ES-CSCs that may be responsible for progression and relapse in some patients (Figure 6G).

### 3.5. Identification of Candidate Drugs for ES-CSC Molecular Targets

Candidate small-molecule inhibitors to 62% (13/21) of the unique molecular targets increased in 3D spheroids from TTC 466 and SK-N-MC cells and ES-CSCs (Figure 7A,B) were identified using our bespoke pipeline (Figure 7C). Of the 279 drugs directed against the targets, 66 (24%) are licensed cancer drugs and 213 (76%) are non-cancer drugs (Figure 7C and Appendix A). The majority of the drugs (202/279; 72%) target one of the two ATP-binding cassette (ABC) drug efflux proteins p-glycoprotein (*n* = 168, includes *n* = 10 targeting p-glycoprotein and additional proteins) or MRP1 (*n* = 13), while 21 drugs target both p-glycoprotein and MRP1. Forty eight of the 279 (17%) drugs hitting the targets have been evaluated in oncology trials, whereas the majority of drugs have been evaluated in non-cancer drug trials (*n* = 986; Figure 7C, Table 2, Appendix A).

POU5F1/OCT-4 interacted with 10/13 of the identified stemness and chemoresistance genes (Figure 7D), suggesting shared functional protein–protein association networks. Five of these 10 gene products (POU5F1/OCT4, c-KIT, CAV1, ITGB1 and CD44) interact (Figure 7D), regulating vital cellular processes (Appendix A). All these proteins are highly expressed in ES-CSCs and so may represent candidate prognostic biomarkers and/or therapeutic targets in ES. We are currently investigating these possibilities. Using our customised pipeline, we identified two FDA approved small-molecule inhibitors, allopurinol and phenytoin, that target POU5F1/OCT4 and are used to treat gout and control seizures in epilepsy, respectively (Table 2, https://www.dgidb.org/genes/POU5F1#_interactions, accessed on 3 January 2023). These two licensed drugs have the highest DGIdb interaction scores of the three compounds that interact with POU5F1/OCT4, inhibit a range of other cancer-relevant molecular targets and are also currently in trials as cancer therapeutics, making them attractive candidate drugs for further in vitro analysis for ES. Several inhibitors targeting multiple receptor tyrosine kinases (TKIs) implicated in the pathogenesis of sarcomas including ES were also identified [52,53], including regorafenib, which is being evaluated in combination with chemotherapy (NCT02085148/2013-003579-36; NCT04055220/REGOSTA; 2021-005061-41/INTER-EWING-1; Table 2).

## 4. Discussion

For the first time, we have identified expression of the membrane-associated ABC transporter protein ABCG1 on the surface of human ES cells. Levels of ABCG1 were increased in SK-N-MC cells forming 3D spheroids compared to cells in 2D culture. Expression of ABCG1 was particularly high in the outer rim of spheroids, suggesting a structural or transport function between cells within and outside the spheroid. Consistent with this hypothesis, knockdown of ABCG1 reduced the ability of ES cells to bind to each other and produce 3D spheroids, reminiscent of the ABCG1-dependent regulation across cellular and intracellular membranes [54]. ABCG1 protein was detected at lower levels in hypoxic cells surrounding the necrotic centre of spheroids, consistent with observations in mouse colon adenocarcinoma spheroids where expression of ABCG1 and hypoxia-inducible factor 1α are inversely correlated [55]. In contrast to studies in lung cancer [56] and normal haemopoietic stem cells [57], decreased ABCG1 had no direct effect on the proliferation or apoptosis of ES cells. Decreased ABCG1 is reported to promote apoptosis through increased expression of the endoplasmic reticulum (ER) stress proteins GRP78 and CHOP [58]. The ability to tolerate high levels of ER stress conferred by the *EWSR1-ETS* oncogene [59] may explain why knockdown of ABCG1 does not induce apoptosis in these ES cells. ABCG1 expression is higher in several cancer types compared to normal tissue [29,56], expression being associated with higher-grade tumours [60], metastasis [61] and poor response to chemotherapy [62]. In agreement with these observations, ABCG1 expression is increased in drug-resistant, self-renewing osteosarcoma cells [18], a second bone cancer characterised by recurrent disease and acquired MDR. Further studies are required to establish what effect this lipid ABC transporter protein has on plasma membrane organisation and recruitment of signalling processes that may regulate cell fate and contribute to tumour development, evasion and metastasis. Single-nucleotide polymorphisms (SNPs) of ABCG1 in the first cytoplasmic domain (within intron 2) are associated with survival of patients with non-small-cell lung cancer [63]. Mutagenesis studies have highlighted the importance of this region for effective trafficking of ABCG1 to the plasma membrane and regulation of cholesterol efflux [64,65]. However, the clinical relevance of the 17 SNPs identified within the cytoplasmic region of ABCG1 (amino acids 178–195, exon 5, www.nvbi.nlm.nih.gov/snp, accessed on 13 July 2022) has yet to be established. In ES 3D spheroids, we identified two novel ABCG1 transcripts, the most abundant transcript (transcript 1) missing exon 5 of the cytoplasmic region. The cellular functional and clinical significance of this deletion and these transcripts requires further investigation. These hypotheses and data are consistent with the premise that ABCG1 identifies ES cells that are capable of surviving chemotherapy and may be responsible for progression and relapse in some patients. We are currently investigating the expression and functional role ABCG1 mRNAs and proteins in ES.

In agreement with the premise that CD133 can identify some cells with a CSC phenotype [14,66], the number of CD133-positive cells and level of CD133 expression was higher in ES cells from 3D spheroids compared to cells grown in 2D. CD133-positive cells shared common characteristics of cancer stem-like cells including increased colony formation from a single cell. However, in patient-derived cells there was no correlation between CD133 expression and self-renewing ability from a single cell. This highlights the difference between established ES cell lines and cultures more recently derived from patient tumours and supports the premise that CD133 does not identify all ES cell populations with a self-renewing phenotype. We are currently investigating the difference in the genotype and phenotype of ES cell lines and patient-derived cells. Although CD133 is reported to be a biomarker of stem cells, this is controversial, which may arise from heterogeneity and uncertainty about its physiological role(s) [67]. In contrast to studies in metastatic melanoma [68], we found no enrichment of the canonical Wnt pathway in CD133-positive ES cells compared to CD133-negative cells [69] and the level of CD133 did not predict outcomes. The increased drug resistance of TC-32 CD133-positive cells compared to CD133-negative cells might instead be effected through greater levels of the drug efflux protein MRP1, overexpression of which induces resistance to chemotherapy [23] and high membrane expression predicts a worse clinical outcome [27]. This suggests the need for further studies on the intrinsic and acquired resistance mechanisms in ES.

Increasing evidence suggests that ES can arise in neurally derived mesenchymal stem cells (MSCs; [70]) or in cells of the neural crest [71]. The interaction between the permissive cellular environment and the EWSR1-ETS tumour-specific chimeric transcription factors leading to cellular transformation and ES is poorly understood, although is no doubt important illustrated by the high incidence of ES in Europeans compared to Africans [72,73,74]. Rewiring of the transcriptome and epigenome, in addition to rare oncogenic driver events including STAG2, TP53 and CDKN2A are also known contributing factors [75]. Interestingly, five of six cell lines produced 3D progeny with projections similar to those displayed by tissue organoids [76], not previously reported in bone cancer. In contrast, the SKES-1 cells generated a cellular core with surrounding dissociating cells reminiscent of ES spheroids from CD133-positive STA-ET 8.2 cells [15]. Heterogeneity of spheroid compactness, growth and stability over time may reflect the amount of extracellular matrix components produced by the different cell lines [77], suggesting these models may be useful tools to evaluate the efficacy of therapeutic candidates. In this study, we found that the ES-CSC phenotype was associated with the expression of ABC transporter proteins and stemness features. However, because of the bystander stemness pathways associated with the cell of origin, unravelling the functional molecular mechanisms leading to Ewing sarcomagenesis requires further investigation.

We identified POU5F1 as a promising candidate therapeutic target, interacting with 10 of the prioritised targets. The protein product of this gene (octamer-binding transcription factor 4, OCT4) regulates several functional characteristics of CSCs, including self-renewal, survival, drug resistance, epithelial–mesenchymal transition and metastasis [78]. This is consistent with emerging roles for this protein in the tumorigenesis of adult cancers and stimulation of expression by the EWSR1 proto-oncogene in several human cancers, including ES [79]. Using our bespoke pipeline, we identified two FDA-approved drugs that are off patent and target POU5F1/OCT4, allopurinol and phenytoin. The efficacy and safety of phenytoin is currently being evaluated in a phase 2 clinical trial as part of a combination maintenance therapy in patients with clinically advanced sarcoma including ES [80] and in patients with metastatic pancreatic cancer [81]. A trial combining allopurinol and mycophenolate mofetil with chemotherapy in patients with relapsed small-cell lung cancer is expected to start recruitment later this year (the CLAMP trial, NCT05049863). We are currently investigating the effect of allopurinol and phenytoin alone and in combination with additional novel drugs and standard of care chemotherapy [82] in ES.

The majority of drugs we identified target the ABC transporter proteins p-glycoprotein and/or MRP1, which contribute to MDR by reducing the level of drug within cells. Several ABC transporter proteins have been described in ES, including MRP1, p-glycoprotein, ABCG1, ABCF1, ABCA6 and ABCA7 [83]. P-glycoprotein has been most frequently studied, although it does not predict outcome [27,84,85,86]. Despite the development of third-generation inhibitors of p-glycoprotein, to date, these drugs are of limited or no clinical value, and the majority of trials have been stopped due to unacceptable toxicity. In contrast, high membrane expression of MRP1 in tumours at diagnosis predicts reduced event-free and overall survival for patients [27]. Interestingly, low levels of ABCF1 in combination with high levels of IGF2BP3 also predicts poor outcome for patients [87]. Despite the clinical failure of inhibitors to ABC transporter proteins, several targeted small molecules, including some TKIs, interact with one or more ABC transporters, suggesting inhibitors may in some cases be beneficial [88]. Increased understanding of the functional role of ABC transporter proteins, including ABCG1, in the cells responsible for progression and relapse is needed to establish which, if any, are worthwhile candidate therapeutic targets to overcome MDR.

## 5. Conclusions

In summary, we have evidence that ABCG1 may be a candidate biomarker that could be used to select ES patients for treatment. This hypothesis requires validation. We have identified proteins expressed by ES-CSCs that might be therapeutic targets and used our bespoke pipeline to identify repurposing drug candidates that have the potential to inhibit these targets and eradicate ES-CSCs. These drugs include FDA-approved small molecules to ABC transporter proteins and two inhibitors of POU5F1/OCT-4.

## Figures and Tables

**Figure 1 cancers-15-00769-f001:**
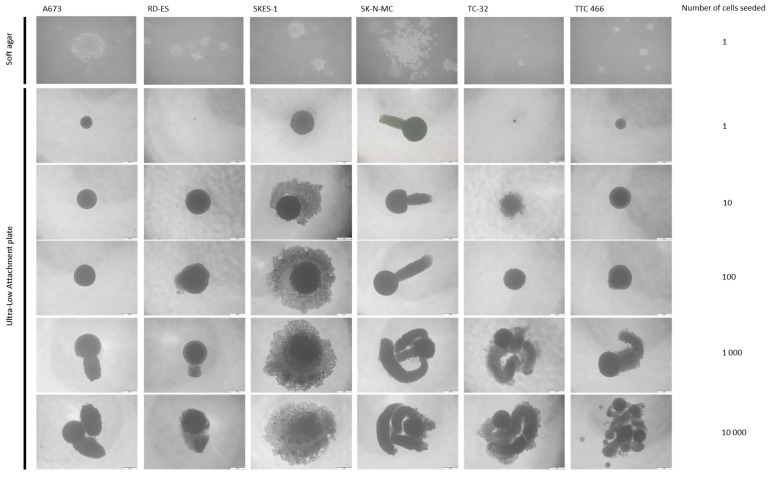
Self-renewing ability of ES cell lines. Single cell suspensions of ES cells were prepared in 0.3% soft agar or 1–10,000 cells seeded into each well of an ultra-low attachment plate (96 wells). Images show colonies formed at 21 days and are representative of three independent experiments.

**Figure 2 cancers-15-00769-f002:**
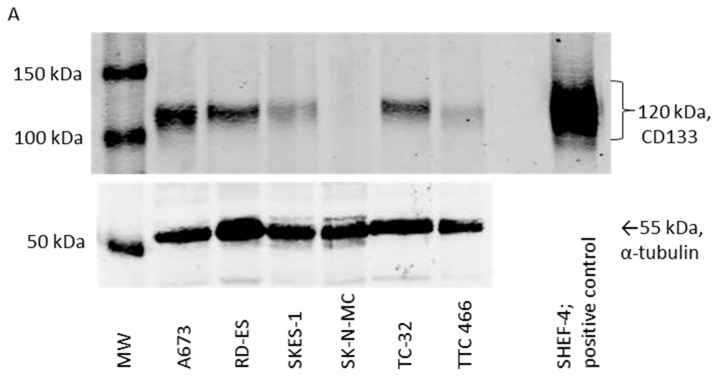
Characterisation of CD133 cells in ES cell lines. (**A**) Protein expression of CD133 in ES cell lines (*n* = 6) was evaluated by Western blot. Equal protein loading was confirmed by probing the blots for α-tubulin. The top differentially expressed genes comparing (**B**) TC-32 and (**C**) A673 CD133-positive and -negative cells using LIMMA were not significant. The difference in ΔCt between the two groups and corresponding Q values are shown. Genes were considered significantly differentially expressed if the Q value was <0.1 and the difference in ∆Ct > 2. Genes with Ct values of >35 in both cell populations were excluded. Dark grey = ABC transporter proteins, grey = pluripotency associated genes, white = Wnt signalling pathway genes. Results of two independent experiments. SHEF-4 embryonic stem cells were included as a positive control for CD133 expression throughout.

**Figure 3 cancers-15-00769-f003:**
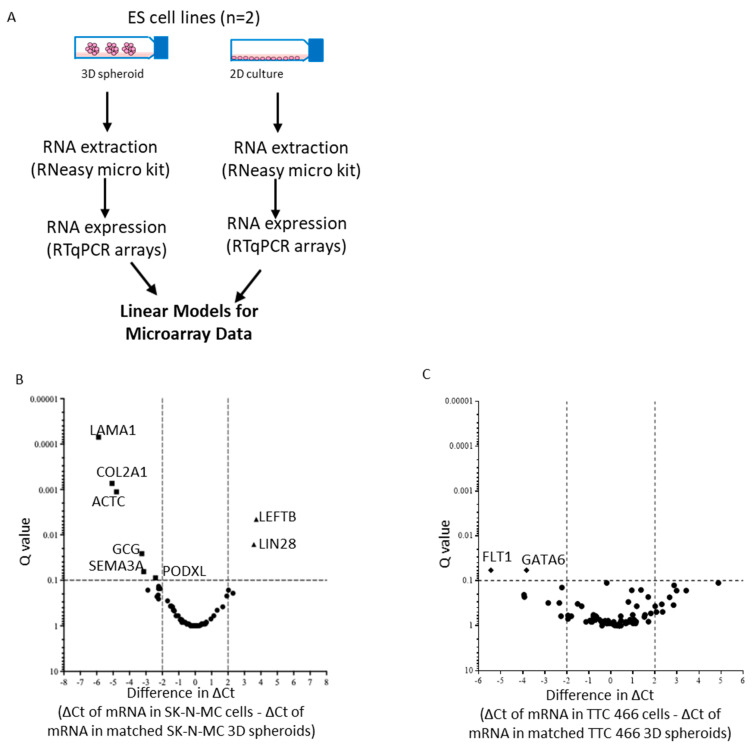
Expression of pluripotency genes in SK-N-MC and TTC 466 cells in 2D and 3D spheroid cultures. (**A**) Summary of strategy to identify shared markers of stemness and MDR in ES cell lines. RNA (1 µg) from (**B**) SK-N-MC and (**C**) TTC 466 cells in 2D and 3D cultures were analysed by RTqPCR using the TaqMan^®^ Human Stem Cell Pluripotency array. The level of mRNA expression is reported using the comparative Ct method, after normalisation of target Ct values to the global mean of all mRNAs. A volcano plot summarising the differentially expressed mRNAs in 2D and 3D spheroids, displaying the Q value (level of significance) and the difference in ΔCt between the cells in 2D and 3D spheroids is shown. The vertical dashed lines indicate the ΔCt thresholds of ±ΔCt of >2 (*x*-axis) and the horizontal line a significant Q value < 0.1 (*y*-axis). Black squares = mRNAs significantly decreased in SK-N-MC 3D spheroids and with a change in ΔCt > 2, Black diamonds = mRNAs significantly decreased in TTC 466 3D spheroids and with a change in ΔCt > 2, black triangles = mRNAs significantly increased in 3D spheroids and with a change in ΔCt >2, black circles = mRNAs with a ΔCt of ±<2 and Q value > 0.1. Genes with Ct values of >35 in 2D cells were excluded. The top differentially expressed genes with a Q value of <0.2 comparing (**D**) SK-N-MC and (**E**) TTC 466 cells in 2D and 3D cultures using LIMMA are shown. The difference in ΔCt between the two groups and corresponding Q values are shown. Genes were considered significantly differentially expressed if the Q value was <0.1 and the difference in ∆Ct > 2.

**Figure 4 cancers-15-00769-f004:**
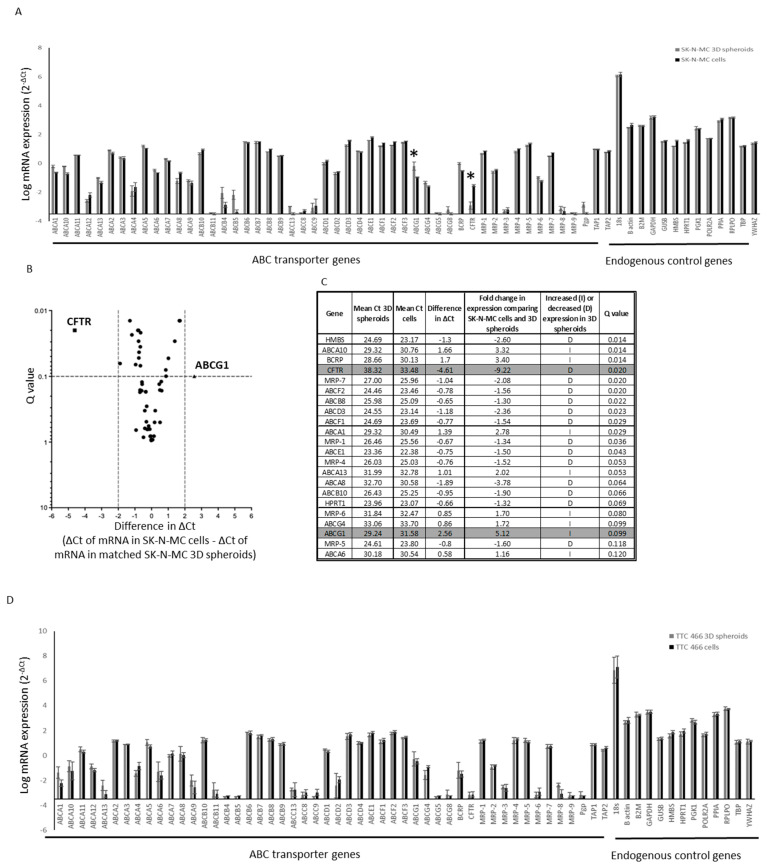
Expression of ABC transporter proteins by SK-N-MC and TTC 466 cells in 2D and 3D spheroid cultures. RNA (1 µg) from (**A**) SK-N-MC and (**D**) TTC 466 cells in 2D and 3D cultures were analysed by RTqPCR using the TaqMan^®^ Human ABC Transporter Array. The level of mRNA expression was reported using the comparative Ct method as mean ± SEM, after normalisation of target Ct values to the global mean of all mRNAs. Results are the mean of two independent experiments. * = Genes with change in ∆Ct > 2 and Q value < 0.1 comparing SK-N-MC cells in 2D and 3D cultures. Genes with Ct values of >35 in SK-N-MC cells grown in 2D were excluded. Volcano plot summarising the differentially expressed ABC transporter mRNAs in (**B**) SK-N-MC and (**E**) TTC 466 2D and 3D spheroids, displaying the Q value (level of significance) and the difference in ΔCt between the cells in 2D and 3D spheroids. The vertical dashed lines indicate the ΔCt thresholds of +/−ΔCt of >2 (*x*-axis) and the horizontal line a significant Q value < 0.1 (*y*-axis). Black square = CFTR which was significantly decreased and black triangle = ABCG1 which was significantly increased in 3D spheroids compared to 2D cultures. Black circles = ABC transporter proteins with a ΔCt of ±<2. The differentially expressed genes all have a Q value < 0.1. Mean Ct values of ABC transporter mRNAs in (**C**) SK-N-MC and (**F**) TTC 466 cells grown in 2D and 3D, the difference in ΔCt between the two groups and corresponding Q values are shown. Grey = genes with change in ∆Ct > 2 and Q value < 0.1. (**G**) ABC transporter and pluripotency mRNAs (*n* = 140) detected (Ct values < 35), highly expressed (Ct values < 25) and not detected (Ct values > 35) in 3D spheroids from TTC 466 and SK-N-MC cells. Venn diagrams show the number of mRNAs not detected, unique or shared for each cell line. (**H**) Validation of ABCG1 mRNA expression by RTqPCR and reported as 2^−ΔΔCt^ in SK-N-MC cells grown in 2D and 3D culture; expression of ABCG1 is normalised to the endogenous control gene PPIA and the control cell line, SHEF-4. RNA expression was compared between 2D and 3D SK-N-MC cells, using a two-tailed *t*-test. The results are representative of 2 independent experiments. (**I**) Increased expression of ABCG1 protein in SK-N-MC cells grown as 3D spheroids compared to 2D cultures was validated by Western blot. Equal loading of each protein was confirmed by probing the Western blot for ß-actin. The results are representative of 2 independent experiments. (**J**) High expression of ABCG1 protein expression in the outer 50 µm region of SK-N-MC spheroids detected by IHC; nuclei are labelled with haematoxylin. Black scale bar = 100 µm. IgG control = SK-N-MC spheroid section incubated with the isotype control antibody (4 μg/mL, Negative Control Mouse IgG1, X0931 (Dako) and 20 μg/μL, Normal Rabbit Serum Control Ig mix, 086199 (Life Technologies), stained with haematoxylin.

**Figure 5 cancers-15-00769-f005:**
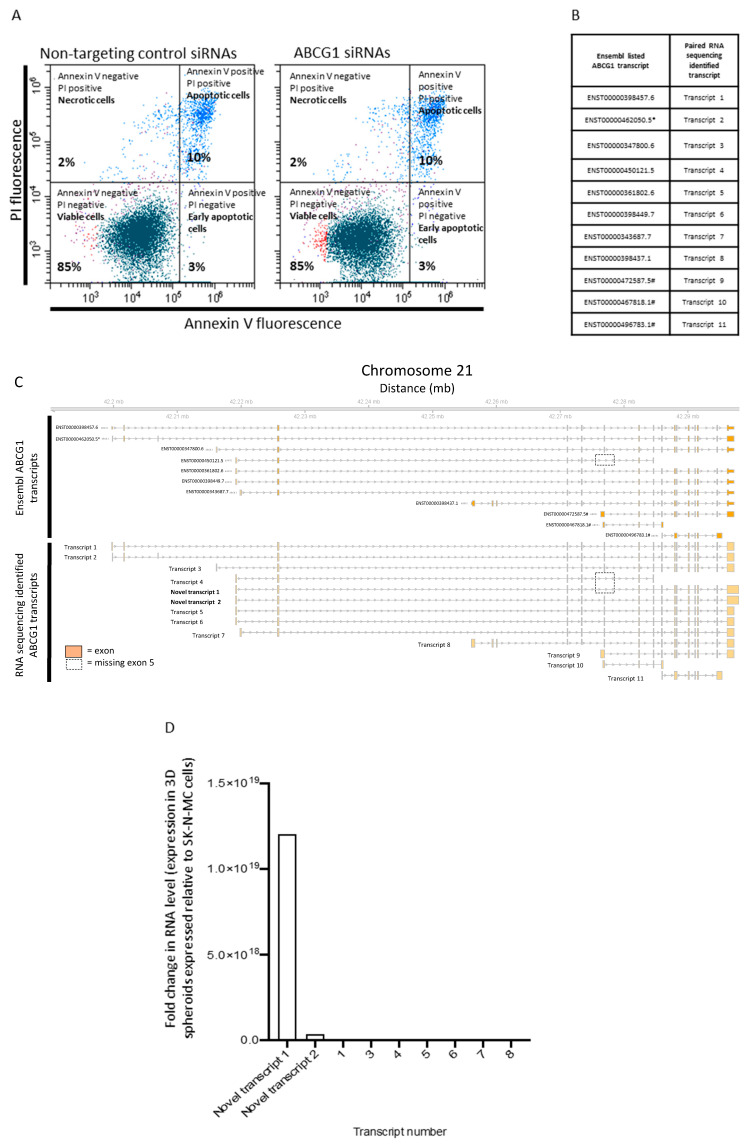
Functional role and characterisation of ABCG1 in ES. (**A**) Representative dot plot of annexin V and PI labelling of SK-N-MC cells analysed by flow cytometry following knockdown of ABCG1 using SMARTpool:Accell ABCG1 siRNA and non-targeting control siRNAs. The upper left quadrant shows the mean percentage of necrotic cells, the upper right = apoptotic cells, lower left = viable cells and the lower right early apoptotic cells. The percentage of apoptotic cells was compared using a non-parametric Mann–Whitney two-tailed *t*-test. (**B**) Canonical ABCG1 RNA transcripts were downloaded from Ensembl.gov and labelled with the prefix ENST and unique transcript number. * = No protein produced from this transcript (www.ensembl.org, accessed on 27 April 2016), # = no protein produced from this transcript, retained intron (www.ensembl.org, accessed on 27 April 2016) (**C**) Canonical transcripts 1 to 11 were detected in SK-N-MC cells in 2D and 3D spheroids. In addition, two novel transcripts (transcript 1 and 2) were identified in 3D spheroids. Orange box = exon, grey arrows = direction of transcription, dotted line box = missing exon 5. (**D**) Novel transcript 1 was the most highly differentially expressed ABCG1 transcript in 3D spheroids compared to cells grown in 2D.

**Figure 6 cancers-15-00769-f006:**
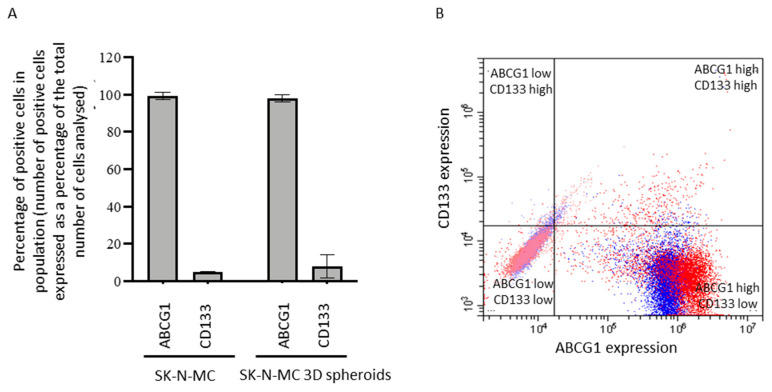
Expression profile of ABCG1, CD133 and MRP1 in ES cell lines and patient-derived cells. (**A**) The percentage of SK-N-MC cells grown in 2D and 3D cultures expressing ABCG1 and CD133 was quantified by flow cytometry; 10,000 events were examined for each condition. The percentage of positive cells is presented as the mean ± SEM (*n* = 6). ABCG1 and CD133 expression in SK-N-MC cells grown in 2D and 3D cultures was compared using a non-parametric Mann–Whitney two-tailed *t*-test. (**B**) Representative dot plot of ABCG1 and CD133 expression of SK-N-MC cells grown in 2D and disaggregated 3D cultures, analysed by flow cytometry. Quadrants represent cells with ABCG1 low CD133 high (**upper left**), ABCG1 high CD133 high (**upper right**), ABCG1 low CD133 low (**lower left)** and ABCG1 high CD133 low (**lower right**) levels of expression. Red spots = cells in 3D culture, blue spots = cells in 2D culture. (**C**) Protein expression of ABCG1 and CD133 in SK-N-MC cells grown in 2D and 3D culture was quantified by flow cytometry. Data are presented as the fold change in fluorescence, expressing the median target protein fluorescence in SK-N-MC cells grown in 3D relative to cells grown in 2D. Results show the mean ± SEM for two independent experiments, 3 replicates per experiment (*n* = 6). ABCG1 and CD133 expression in 2D and 3D cultures was compared using an unpaired two-tailed *t*-test. (**D**) ABCG1 protein expression in CD133-positive and CD133-negative TC-32 cells detected by Western blot. Equal protein loading was confirmed by probing the blots for ß-actin. Results are representative of 2 independent experiments. (**E**) ABCG1 protein expression in patient-derived ES cells, determined by Western blot. Equal protein loading was confirmed by probing the blots for ß-actin. Results are representative of 2 independent experiments. (**F**) There was no correlation between the level of CD133 expression and ability to produce progeny from a single cell. The level of CD133 expression was quantified by flow cytometry and the ability to produce progeny from a single cell evaluated in 2D adherent culture at 21 days. Correlation was examined using linear regression. Open circles = cell lines, filled circles = patient-derived cells. (**G**) Summary of ABCG1 and MRP1 protein expression in ES cells grown in 2D and 3D culture. Salmon-pink circle = ES cells grown in 2D, orange circle = ES cells grown in 3D (spheroids), blue circle = ABCG1 protein, green circle = CD133 protein, black line = plasma membrane.

**Figure 7 cancers-15-00769-f007:**
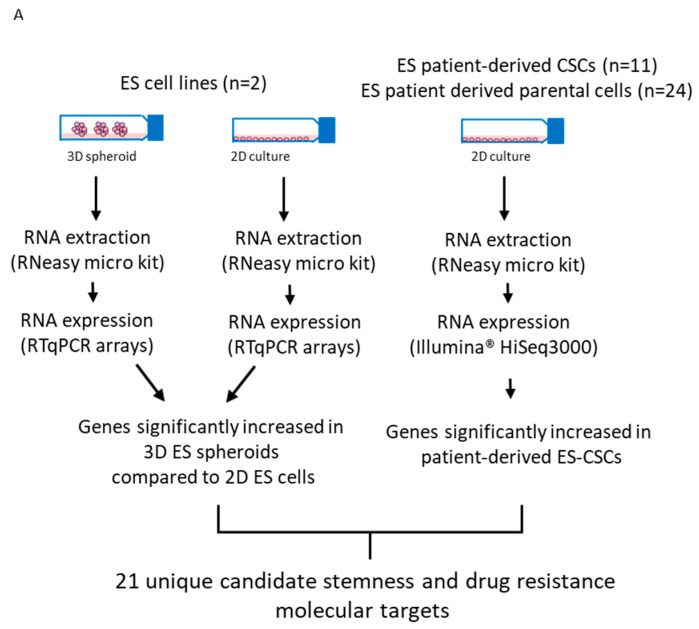
Identification of candidate shared stemness and chemoresistance therapeutic targets. (**A**) Summary of strategy to identify shared markers of stemness and MDR in ES preclinical models. (**B**) Stemness and MDR-associated genes increased in 3D spheroids and patient-derived ES-CSCs. The full and alternative gene names, the Ensembl gene abbreviation (Gene) and the data from which genes were identified (Source of the gene) are shown. The source of the gene is presented as 1 or 2, where 1 = target gene identified from transcriptome analysis of patient-derived ES-CSCs [17] and 2 = target gene identified by comparing transcriptome of SK-N-MC and TTC 466 cells grown in 3D and 2D (Figure 3 and Figure 4). (**C**) Pipeline to identify candidate drugs based on 21 stemness and chemoresistance genes. Green ovals = drug-repurposing data sources, pink ovals = public data sources, blue squares = number of targets or drugs at each step. (**D**) Predicted interaction between the 13 shared stemness and chemoresistance genes (excluding p-glycoprotein and MRP1) that have corresponding candidate drugs using STRING [30]. Five of the genes are known to bind and regulate vital biological processes, and could be candidate prognostic biomarkers and/or therapeutic targets in ES. Pink lines = known interaction that has been experimentally determined, extracted from BIND, DIP, GRID, HPRD, IntAct, MINT and PID databases. Blue lines = known interaction based on data from curated databases Biocarta, BioCyc, GO, KEGG and Reactome. Grey lines = proteins reported to be co-expressed. Green lines = interactions identified by text mining. Filled nodes = 3D structure known or predicted. Coloured nodes = first shell of interacting proteins. White nodes = second shell of interacting proteins. Parental cells = patient-derived bulk population of ES cells from which the ES-CSCs were derived [14].

**Table 1 cancers-15-00769-t001:** Self-renewing ability of ES cell lines. A single cell suspension of each cell line was seeded into soft agar or into a 96 well ultra-low adherence plate at a density of 1–10,000 cells. The number of colonies was counted after 21 days. For studies in soft agar, the mean (±SEM) percentage colony formation is expressed as the number of colonies counted at 21 days relative to the number of cells seeded. For cells seeded onto an ultra-low adherence plate, the mean (±SEM) percentage colony formation is expressed as the number of wells containing a colony of >5 cells relative to the total number of wells with cells seeded. Results are representative of three independent experiments. Percentage colony formation was compared using ANOVA and Bonferroni’s post hoc test.

Cell Line	Clone or Spheroid Formation (%)					
	Clone Formation in Soft Agar from a Single Cell	Ultra-Low Attachment Plate				
		1 Cell	10 Cells	100 Cells	1000 Cells	10,000 Cells
A673	17 ±1	18 ± 5	88 ± 10	100	100	100
RD-ES	17 ±1	5 ± 1	45 ± 13	100	100	100
SKES-1	29 ± 2 (*p* < 0.05)	75 ± 7 (*p* < 0.05)	98 ± 2	100	100	100
SK-N-MC	12 ± 1	64 ± 1	97 ± 3	100	100	100
TC-32	7 ± 0.5 (*p* < 0.05)	15 ± 3	93 ± 9	100	100	100
TTC 466	13 ± 1	45 ± 2	96 ± 6	100	100	100

**Table 2 cancers-15-00769-t002:** Drugs directed against the shared stemness and chemoresistance therapeutic targets. Drugs directed against the shared ES stemness and chemoresistance targets that have been used in human clinical trials, identified using our in-house pipeline. Drug names, direct molecular targets, whether the drug has been previously used in the treatment of cancer patients and is on- or off-patent are shown. The number, trial phase and status, cancer type and detailed trial information of clinical trials are listed. NA = no phase 3 or phase 4 trials recorded for the drug. BNFC = British National Formulary- Children, a directory or all drugs approved for use in children. Y = yes, N = no. * = direct ES targets of fostamatinib (KIT, STK10, SRC, SIK1, RPS6KA1, ROCK2, RET, PTK2B, PTK2, PRKG2, PRKCD, PLK4, PLK3, PLK1, PIM3, PIK3CG, PIK3CD, PDGFRB, PDGFRA, PAK3, PAK1, NTRK3, NTRK2, NTRK1, NEK2, MST1R, MAPK14, MAP3K3, MAP2K2, LYN, LIMK1, KDR, ITK, INSRR, INSR, GSK3A, MTOR, FLT4, FLT3, FLT1, FGFR1, ERN1, ERBB4, ERBB2, EPHB4, EPHB2, EPHA3, EPHA2, EIF2AK4, EGFR, DYRK1B, DCLK3, DCLK1, CSK, CSF1R, CHEK2, CHEK1, CDK4, BTK, BRAF, AXL, AURKB, AURKA, ALK, ZAP70, WEE1, TNK2, TNK1, TIE1, TGFBR2, TEK, TAOK3, STK36, STK33, MET, JAK1, ABL1, SYK).

Drug	Ewings Sarcoma Targets	Cancer Drug	BNFC approval	Off-Patent	Trial Count	Max Trial Phase	Trial Phase (Number of Trials in Phase)	Trial Status (Number of Trials in Status)	Cancer Type (Number of Trials in Cancer Type)	Trial Information for Phase 3 and Phase 4 trials (Trial Number, Phase, Status, Recruitment Status, Cancer Type)
Cabozantinib	KIT, RET, KDR, MET	Y	N	N	6	Phase 2	Phase 1 (1), Phase 2 (5)	Recruiting (1), Active, not recruiting (3), Not yet recruiting (1), Other (1)	Multiple sarcomas, including Ewing’s (1), Mixed paediatric, including Ewing’s (3), Ewing’s or osteosarcoma (2)	NA
Dasatinib	KIT, MAPK14, LYN, HSPA8, EPHB4, CSK, BCR, BTK, PDGFRB, EPHA2, SRC, ABL1	Y	Y	Y	2	Phase 2	Phase 1/2 (1), Phase 2 (1)	Active, not recruiting (1), Completed (1)	Mixed paediatric, including Ewing’s (1), Multiple sarcomas, including Ewing’s (1)	NA
Decitabine	DNMT3B, DNMT1	Y	N	Y	1	Phase 1	Phase 1 (1)	Completed (1)	Mixed, including sarcomas (1)	NA
Erdafitinib	KIT, KDR, PDGFRB, PDGFRA, CSF1R, RET, FGFR1	Y	N	N	1	Phase 2	Phase 2 (1)	Recruiting (1)	Mixed paediatric, including Ewing’s (1)	NA
Imatinib	KIT, ABCB1, PDGFRB, ABL1, PDGFRA, CSF1R, NTRK1, RET, BCR	Y	Y	Y	4	Phase 2	Phase 2 (4)	Completed (4)	Mixed, including sarcomas (1), Multiple sarcomas, including Ewing’s (1), Mixed paediatric, including Ewing’s (1), Ewing’s or DSRCT (1)	NI
Lenvatinib	KIT, RET, PDGFRA, FGFR1, FLT4, KDR, FLT1	Y	N	N	1	Phase 1/2	Phase 1/2 (1)	Active, not recruiting (1)	Mixed paediatric, including Ewing’s (1)	NA
Pazopanib	KIT, ITK, PDGFRB, PDGFRA, FLT4, KDR, FLT1	Y	N	Y	1	Phase 2	Phase 2 (1)	Completed (1)	Mixed paediatric, including Ewing’s (1)	NA
Regorafenib	KIT, RET, ABL1, BRAF, EPHA2, NTRK1, TEK, FGFR1, PDGFRB, PDGFRA, FLT4, KDR, FLT1	Y	N	N	5	Phase 2	Phase 1 (1), Phase 1/2 (1), Phase 2 (2), Other (1)	Recruiting (4), Active, not recruiting (1)	Multiple sarcomas, including Ewing’s (3), Ewing’s or osteosarcoma (1), Mixed paediatric, including Ewing’s (1)	NA
Sorafenib	KIT, ABCB1, FLT1, RET, FGFR1, PDGFRB, FLT3, KDR, FLT4, BRAF	Y	N	Unclear	2	Phase 2	Phase 1 (1), Phase 2 (1)	Active, not recruiting (1), Completed (1)	Ewing’s or DSRCT (1), Mixed paediatric, including Ewing’s (1)	NA
Sunitinib	KIT, ABCB1, PDGFRA, CSF1R, FLT3, FLT4, KDR, FLT1, PDGFRB	Y	N	Y	2	Phase 2	Phase 1/2 (1), Phase 2 (1)	Ongoing (1), Completed (1)	Multiple sarcomas, including Ewing’s (1), Mixed, including sarcomas (1)	NA
Temozolomide	ABCB1	Y	Y	Y	23	Phase 3	Phase 1 (10), Phase 1/2 (4), Phase 2 (8), Phase 3 (1)	Recruiting (8), Active, not recruiting (4), Completed (6), Not yet recruiting (2), Other (2), Terminated (1)	Mixed paediatric, including Ewing’s (6), Ewing’s (10), Mixed, including sarcomas (4), Ewing’s or DSRCT (1), Ewing’s or RMS (2)	NCT03495921 (Phase 3; Active, not recruiting)—Ewing sarcoma
Aliskiren	ABCB1	N	N	Y	1	Phase 2	Phase 2 (1)	Recruiting (1)	Multiple cancer types (1)	NA
Allopurinol	POU5F1	N	Y	Y	1	Phase 1/2	Phase 1/2 (1)	Not yet recruiting (1)	Lung (1)	NA
Atorvastatin	ABCB1, NR1I3, HDAC2, AHR, DPP4, HMGCR	N	Y	Y	16	Phase 3	Phase 1 (4), Phase 2 (6), Phase 2/3 (1), Phase 3 (5)	Recruiting (13), Ongoing (1), Not yet recruiting (2)	Multiple cancer types, Leukaemia (1), GI (4), Breast (8), Urological (3)	NCT03024684 (Phase 3; Recruiting)—Hepatocellular carcinoma, NCT03819101 (Phase 3; Recruiting)—Prostate, NCT03971019 (Phase 3; Recruiting)—Breast, NCT04601116 (Phase 3; Recruiting)—Breast, NCT04026230 (Phase 3; Recruiting)—Prostate
Azithromycin	ABCC1	N	Y	Y	3	Phase 2	Phase 2 (3)	Ongoing (3)	Lymphoma (2), Breast (1)	NA
Carvedilol	ABCB1, HIF1A, GJA1, VEGFA	N	Y	Y	2	Phase 2	Phase 1 (1), Phase 2 (1)	Active, not recruiting (1), Not yet recruiting (1)	Urological (1), CNS (1)	NA
Chlorpromazine	ABCB1	N	N	Y	3	Phase 2	Phase 1 (1), Phase 1/2 (1), Phase 2 (1)	Recruiting (2), Not yet recruiting (1)	GI (1), CNS (2)	NA
Citalopram	ABCB1	N	Y	Y	1	Phase 3	Phase 3 (1)	Ongoing (1)	CNS (1)	2013-004705-59 (Phase 3; Ongoing)—Glioblastoma
Clarithromycin	ABCB1	N	Y	Y	18	Phase 4	Phase 1/2 (2), Phase 2 (10), Phase 3 (3), Phase 4 (1), Other (2)	Recruiting (8), Active, not recruiting (8), Not yet recruiting (1), Other (1)	Lymphoma (1), Other Haem-onc (15), Multiple cancer types (1), GI (1)	ChiCTR2100047608 (Phase 4; Recruiting)—Multiple Myeloma, NCT02575144 (Phase 3; Active, not recruiting)—Multiple Myeloma, NCT02516696 (Phase 3; Active, not recruiting)—Multiple Myeloma, NCT04287660 (Phase 3; Recruiting)—Multiple Myeloma
Clopidogrel	ABCB1	N	N	Y	1	Phase 1	Phase 1 (1)	Recruiting (1)	Head and Neck (1)	NA
Colchicine	ABCB1, TUBB	N	N	Y	2	Phase 2	Phase 1 (1), Phase 2 (1)	Recruiting (2)	Urological, Multiple cancer types (1), GI (1)	NA
Cyclosporine	ABCC1, ABCB1	N	N	Y	2	Phase 2	Phase 1 (1), Phase 2 (1)	Recruiting (1), Active, not recruiting (1)	Leukaemia (1), Other Haem-onc (1)	NA
Deferoxamine	ABCB1	N	N	Y	4	Phase 3	Phase 1 (2), Phase 2 (1), Phase 3 (1)	Recruiting (3), Not yet recruiting (1)	Multiple cancer types (1), Breast (1), Leukaemia (1), GI (1)	IRCT20200313046756N2 (Phase 3; Pending)—Acute Myeloid Leukaemia
Digoxin	ABCB1	N	Y	Y	2	Phase 2	Phase 1 (1), Phase 2 (1)	Recruiting (2)	GI (1), Multiple cancer types (1)	NA
Disulfiram	ABCB1	N	N	Y	13	Phase 2/3	Phase 1 (4), Phase 1/2 (2), Phase 2 (6), Phase 2/3 (1)	Recruiting (7), Active, not recruiting (2), Ongoing (3), Not yet recruiting (1)	Multiple cancer types, GI (1), Soft-Tissue Sarcoma, Bone Sarcoma (1), CNS (3), Other Haem-onc (1), Urological (2), Multiple cancer types, Breast (1), GI (2), Breast (2)	NA
Doxycycline	ABCB1	N	Y	Y	10	Phase 2	Phase 1 (1), Phase 2 (9)	Recruiting (4), Active, not recruiting (4), Ongoing (1), Not yet recruiting (1)	Urological (1), Lymphoma (3), Multiple cancer types (1), Breast, Gynaecological (1), Breast (1), GI (2), Head and Neck (1)	NA
Fenofibrate	ABCB1, NR1I2	N	Y	Y	1	Phase 2	Phase 2 (1)	Recruiting (1)	CNS (1)	NA
Fostamatinib	*	N	N	N	2	Phase 1	Phase 1 (2)	Recruiting (2)	Gynaecological (1), Other Haem-onc, Leukaemia (1)	NA
Indomethacin	ABCC1, PTGS2	N	Y	Y	5	Phase 4	Phase 1 (1), Phase 1/2 (1), Phase 2 (1), Phase 3 (1), Phase 4 (1)	Recruiting (2), Active, not recruiting (3)	Urological (2), Breast (1), Head and Neck (2)	ChiCTR2000038968 (Phase 4; Recruiting)—Prostate, NCT01265849 (Phase 3; Active, not recruiting)—Cancer of the oral cavity
Itraconazole	ABCB1, ERBB2	N	Y	Y	15	Phase 3	Phase 1 (4), Phase 1/2 (2), Phase 2 (7), Phase 3 (1), Other (1)	Recruiting (9), Active, not recruiting (2), Completed (1), Not yet recruiting (1), Other (1), Suspended (1)	Leukaemia, Other Haem-onc (1), Lung (2), Multiple cancer types, Lymphoma, Leukaemia (1), follow-up continuing (1), Multiple cancer types (2), GI (4), Urological (1), Gynaecological (2), Skin (1)	NCT03458221 (Phase 3; Not yet recruiting)—Ovarian
Ivermectin	ABCB1	N	Y	Y	2	Phase 2	Phase 2 (2)	Recruiting (1), Not yet recruiting (1)	Breast (1), Multiple cancer types (1)	NA
Ketoconazole	ABCB1, NR1I3, NR1I2, AR	N	Y	Y	4	Phase 2	Phase 1 (2), Phase 2 (2)	Recruiting (1), Active, not recruiting (2), Ongoing (1)	Urological (2), Breast, CNS (1), CNS (1)	NA
Lansoprazole	ABCB1, MAPT	N	Y	Y	3	Phase 3	Phase 2 (1), Phase 3 (2)	Recruiting (2), Active, not recruiting (1)	Breast (1), Lymphoma (1), GI (1)	NCT04874935 (Phase 3; Recruiting)—Breast, NCT03647072 (Phase 3; Recruiting)—Non-Hodgkin Lymphoma
Levetiracetam	ABCB1	N	Y	Y	2	Phase 2	Phase 2 (1), Other (1)	Not yet recruiting (2)	CNS (2)	NA
Losartan	ABCB1	N	N	Y	12	Phase 3	Phase 1 (4), Phase 2 (7), Phase 3 (1)	Recruiting (9), Active, not recruiting (1), Not yet recruiting (2)	Multiple cancer types (3), Bone Sarcoma (1), GI (7), Breast (1)	CTRI/2021/05/033482 (Phase 3; Not Yet Recruiting)—Pancreatic
Lovastatin	ABCB1, HDAC2, HMGCR	N	N	Y	1	Other	Other (1)	Not yet recruiting (1)	GI (1)	NA
Maprotiline	ABCB1	N	N	Y	1	Phase 1	Phase 1 (1)	Not yet recruiting (1)	CNS (1)	NA
Mefloquine	ABCB1	N	Y	Y	1	Phase 1	Phase 1 (1)	Active, not recruiting (1)	CNS (1)	NA
Miconazole	ABCB1, NR1I2	N	Y	Y	2	Phase 2	Phase 1 (1), Phase 2 (1)	Recruiting (2)	Multiple cancer types (2)	NA
Midazolam	GABRB3, ABCB1	N	Y	Y	2	Phase 2	Phase 2 (2)	Recruiting (1), Not yet recruiting (1)	Urological (2)	NA
Mifepristone	ABCB1, NR1I2, KLK3, NR3C1, PGR	N	N	Y	2	Phase 3	Phase 2 (1), Phase 3 (1)	Active, not recruiting (1), Not yet recruiting (1)	Breast (2)	NCT05016349 (Phase 3; Not yet recruiting)—Breast
Miltefosine	ABCB1	N	N	Y	2	Phase 2	Phase 2 (2)	Recruiting (1), Active, not recruiting (1)	Breast (1), Multiple cancer types (1)	NA
Nelfinavir	ABCB1	N	N	Y	9	Phase 3	Phase 1 (2), Phase 1/2 (2), Phase 2 (3), Phase 3 (2)	Recruiting (5), Active, not recruiting (2), Ongoing (2)	Soft Tissue Sarcoma (1), Urological, Multiple cancer types (1), Gynaecological (3), Head and Neck (1), Other Haem-onc (1), GI (1), CNS (1)	NCT03256916 (Phase 3; Recruiting)—Advanced carcinoma of the cervix, CTRI/2017/08/009265 (Phase 3; Open to Recruitment)—Advanced carcinoma of the cervix
Nicardipine	ABCB1	N	N	Y	1	Other	Other (1)	Not yet recruiting (1)	CNS (1)	NA
Omeprazole	ABCB1, AHR	N	Y	Y	3	Phase 2	Phase 1 (2), Phase 2 (1)	Recruiting (2), Active, not recruiting (1)	Urological (1), Breast (1), GI (1)	NA
Pantoprazole	ABCB1	N	N	Y	2	Phase 2	Phase 1/2 (1), Phase 2 (1)	Active, not recruiting (1), Not yet recruiting (1)	Head and Neck (1), Urological (1)	NA
Phenytoin	POU5F1, SCN8A, SCN2A, SCN3A, NR1I2, SCN1A, SCN5A	N	Y	Y	1	Phase 2	Phase 2 (1)	Recruiting (1)	Bone Sarcoma, Soft-Tissue Sarcoma (1)	NA
Pravastatin	ABCB1, HDAC2, HMGCR	N	N	Y	2	Phase 4	Phase 2 (1), Phase 4 (1)	Active, not recruiting (1), Not yet recruiting (1)	Leukaemia (1), Breast (1)	ChiCTR2000034035 (Phase 4; Pending)—Breast
Propofol	GABRB3, ABCB1, SCN2A, SCN4A	N	Y	Y	29	Phase 4	Phase 2 (1), Phase 3 (2), Phase 4 (7), Other (19)	Recruiting (17), Active, not recruiting (2), Ongoing (1), Not yet recruiting (9)	GI (6), Breast, GI (1), Multiple cancer types (7), Breast (5), Urological (3), Lung (6), CNS (1)	NCT01975064 (Phase 4; Recruiting)—Breast, colon, rectal, NCT05331911 (Phase 4; Not yet recruiting)—Liver, NCT04475705 (Phase 4; Recruiting)—Paediatric solid tumours, NCT05141877 (Phase 4; Not yet recruiting)—Brain tumour, NCT03034096 (Phase 4; Recruiting)—Adult cancer, NCT04513808 (Phase 3; Recruiting)—Oesophageal caner, ChiCTR2000040604 (Phase 4; Pending)—Non-small cell lung cancer, ACTRN12611000301965 (Phase 4; Not yet recruiting)—Breast, 2009-009114-40 (Phase 3; Ongoing)—Prostate
Propranolol	ABCB1, EGFR, ADRB3	N	N	Y	19	Phase 3	Phase 1 (3), Phase 1/2 (1), Phase 2 (13), Phase 2/3 (1), Phase 3 (1)	Recruiting (10), Active, not recruiting (1), Not yet recruiting (7), Other (1)	GI (8), Urological (2), Soft Tissue Sarcoma (2), Gynaecological (1), Skin (3), Other (1), Multiple cancer types (2)	CTRI/2019/11/021924 (Phase 3; Not Yet Recruiting)—Ovarian
Rifampicin	ABCB1, NR1I2	N	Y	Y	1	Phase 1	Phase 1 (1)	Recruiting (1)	Multiple cancer types, Lymphoma, Leukaemia (1)	NA
Ritonavir	ABCC1, ABCB1, NR1I2	N	Y	Y	1	Phase 1	Phase 1 (1)	Recruiting (1)	Leukaemia, Lymphoma (1)	NA
Sertraline	ABCB1, SLC29A4, SLC6A3	N	Y	Y	1	Phase 1	Phase 1 (1)	Recruiting (1)	Leukaemia (1)	NA
Simvastatin	ABCB1, HDAC2, HMGCR	N	Y	Y	18	Phase 4	Phase 1 (4), Phase 2 (11), Phase 3 (1), Phase 4 (2)	Recruiting (7), Active, not recruiting (3), Ongoing (3), Not yet recruiting (4), Suspended (1)	Breast (5), Other Haem-onc (1), Lung (3), Gynaecological (1), GI (6), Multiple cancer types (2)	ChiCTR2000034035 (Phase 4; Pending)—Breast, 2010-018491-24 (Phase 4; Ongoing)—Adults with bone metastasis, NCT03971019 (Phase 3; Recruiting)—Breast
Sirolimus	ABCB1, MTOR	N	Y	Y	28	Phase 4	Phase 1 (7), Phase 1/2 (4), Phase 2 (13), Phase 3 (2), Phase 4 (1), Other (1)	Recruiting (17), Active, not recruiting (7), Ongoing (3), Not yet recruiting (1)	Lung (2), Multiple cancer types, Lung (1), Multiple cancer types (6), Other (2), Soft-Tissue Sarcoma (3), Urological, Multiple cancer types (1), Bone Sarcoma, Soft Tissue Sarcoma (1), Gynaecological (1), Leukaemia (3), Endocrine (1), GI (3), CNS (1), Bone Sarcoma (2), Breast (1)	NCT04775173 (Phase 3; Recruiting)—Haemangioendothelioma, ChiCTR1900021896 (Phase 4; Recruiting)—Liver, NCT04736589 (Phase 3; Not yet recruiting)—Breast
Valganciclovir	ABCB1	N	Y	Y	5	Phase 2	Phase 1/2 (2), Phase 2 (2), Other (1)	Recruiting (4), Not yet recruiting (1)	GI (1), Lymphoma (1), Head and Neck (1), CNS (1), Multiple cancer types (1)	NA
Verapamil	ABCC1, ABCB1	N	N	Y	1	Phase 1	Phase 1 (1)	Active, not recruiting (1)	Lymphoma (1)	NA
Warfarin	ABCB1, AXL, NR1I2	N	N	Y	1	Phase 1	Phase 1 (1)	Recruiting (1)	GI (1)	NA
Zidovudine	ABCC1, ABCB1, TERT	N	Y	Y	1	Phase 2	Phase 2 (1)	Recruiting (1)	Leukaemia (1)	NA

## Data Availability

The FASTQ files of sequenced ES cell lines, 2D and 3D SK-N-MC cells are available in the Research Data Leeds Repository (University of Leeds), Burchill Susan and Roundhill Elizabeth (2022): Total RNA sequencing of ES cell lines, TTC 466 and SK-N-MC 2D and 3D cultures. [Dataset]. https://doi.org/10.5518/1210.

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
