# Peer review of "Exploiting the Stemness and Chemoresistance Transcriptome of Ewing Sarcoma to Identify Candidate Therapeutic Targets and Drug-Repurposing Candidates"

_cancers, 2023, doi:10.3390/cancers15030769_

Round 1
Reviewer 1 Report
This manuscript presented by Roundhill et al. investigates a potential role of CD133, ABCG1 for maintenance cancer stem-like cells in Ewing sarcoma. Additionally, by analyzing gene expression profiles in 2D culture and 3D spheroids, they identified potential 21 stemness-associated genes, among which POU5F1/OCT4 is proposed to be a prominent target which could be targeted by repurposing agents, allopurinol and phenytoin.
Although this manuscript investigates scientifically and clinically important aspects, characterization of stemness and its therapeutic implementation in Ewing sarcoma, the major concern is that the data presented by the authors are largely cell line-dependent, thus inconsistent, unable to draw authors’ conclusions, which are largely based on their assumption. Additionally, the methods are frequently provided only by citations and described insufficiently to reproduce the procedures.
The major conclusion stated in the abstract “ABCG1, a potential oncogene in some cancers, was overexpressed in ES CSCs where it may play a role in plasma membrane organisation and drug resistance.” is not supported by the data, rather based on the assumption by the authors. Likewise, the statement “In summary, we have identified a novel marker of ES-CSCs and licensed cancer and non-cancer drugs to targets expressed by ES-CSCs that may be repurposed for evaluation in ES” cannot be supported since the data are again inconsistent, thus inconclusive. Critically, the validation of drug efficacy and specificity to ES-CSCs, which is necessary to draw authors’ conclusions, is lacking.
Accordingly, this manuscript seems to be too preliminary to this Reviewer and it is not recommended for publication of this manuscript with current authors’ conclusions.
Specific comments
Major concerns
1. Line number 238
Since CD133 expression is not associated with self-renewing phenotype and is not necessarily associated with drug resistant (cf. TC-32 and A673) as the authors demonstrate, the statement “CD133 as a marker of self-renewing drug resistant ES-CSCs” is entirely incorrect.
2. Line numbers 243-245
The authors state “However, there was no difference in proliferation, cell cycle status or telomere length, phenotypes frequently associated with CSCs and self-renewing ability”, which does not support the notion that CD133 is an appropriate marker for self-renewing drug resistant ES-CSCs.
3. Line numbers 250-256
The authors state “Consistent with the cellular origin of ES and the high levels of stemness markers, no significant differentially expressed genes associated with pluripotency, ABC transporter and Wnt signalling pathways were identified in CD133-positive and CD133-negative cells). These observations are also consistent with the premise that CD133 is a marker of ES-CSCs, although pathways classically associated with the CSC phenotype in other cancer types are not differentially expressed in CD133-positive and CD133-negative ES cells.” The data showing that the gene signatures of CD133-positive cells are not associated with stemness features again indicate that CD133 is not an appropriate marker for self-renewing drug resistant ES-CSCs.
4. Line number 270
To this Reviewer it is unclear why low or no detectable CD133 protein is the reason for comparing the transcriptome of TTC 466 and SK-N-MC in 3D spheroids and 2D cultures.
5. Line numbers 348-349
If the authors used a non-parametric statistic test such as Mann-Whitney test with two biologically independent experiments, due to the lower statistical power, it is impossible to give a result of P=0.0015.
6. Line numbers 408-413
Again, these results are additional evidence that CD133 is not an appropriate marker for self-renewing CSCs also for patient-derived ES cells.
7. Line numbers 448-449
Although there is no reliable marker for CSCs, the authors designate ES patient-derived CSCs in Figure 7A. Since the clear definition of putative CSCs is essential for the data interpretation, the authors should provide methodological details.
8. Line number 483
The authors need to provide information why these proteins are candidate prognostic biomarkers and/or therapeutic targets in ES.
9. Line number 485
The authors need to cite adequate references on drug specificity to POU5F1/OCT4.
10. Line numbers 9-10 in Discussion
Low ABCG1 protein signals can still be observed in Figure 4J based on the negative IgG control. The authors should correct the statement accordingly.
11. Line numbers 64-68 in Discussion
The authors do not provide sufficient evidence for the statement “In this study we found the ES-CSC phenotype was most strongly associated with expression and functional activity of ABC transporter proteins rather than stemness features or secondary genetic events”.
12. Line numbers 103-104 in Conclusions
The authors do not provide sufficient evidence to conclude that “ABCG1 as a candidate biomarker to select ES patients for treatment”
Minor concerns
1. Line number 49
To this Reviewer it is not sufficient to state that “ES is a cancer stem-cell driven malignancy” from the cited references.
2. Line numbers 145-147
The authors need to provide an argument why expression thresholds were set at 25 and 35.
3. Line number 221
To this Reviewer the word “histology” seems not appropriate since spheroids do not contain any tissue structure.
4. Line number 224
The authors describe 3D projections as a biologically relevant structure. However, it is unclear if the structure would be related with any biological significance or rather caused by experimental settings without more detailed description.
5. Table 2 is hardly to read in the manuscript, it should not be integrated in the manuscript.
6. Line numbers 15-16 in Discussion
The authors should cite appropriate references for this statement.
7. Line numbers 20-22 in Discussion
Based on the Supplementary Data 6 high ABCG1 expression is statistically significantly associated with better patient survival. Accordingly, the statement “These observations are consistent with the premise that ABCG1 identifies ES cells that are capable of surviving chemotherapy and may be responsible for progression and relapse in some patients.” is too ambiguous to claim a potential role of ABCG1 for clinical outcome. The authors need to discuss more detailed on the basis of the Supplementary Data 6.
8. Line numbers 54-56 in Discussion
To this Reviewer the sentences do not make any sense.
9. The overall scientific English level is acceptable.
Reviewer 2 Report
Kindly send the manuscript again with the updated figure. The majority of the figures in this manuscript are not visible.
Thanks
Author Response
The updated version has been submitted. All figures are in the main text.
Round 2
Reviewer 1 Report
The authors have improved the manuscript by correcting and modifying sentences and statements according to the suggestions by this Reviewer. However, there are still major and minor concerns especially about the drug specificity for allopurinol and phenytoin, which the authors did not clearly solve in the revised manuscript. This point is very important to avoid any misinterpretation by readers that allopurinol and phenytoin would significantly interact with POU5F1/OCT4 proteins, for which the authors did not provide detailed information how the inhibition of POU5F1/OCT4 would be realistic from the viewpoint of clinical translation. Furthermore, there is a concern about the antibody specificity for ABCG1. The validation of the antibody specificity is fundamental to draw a scientifically robust conclusion.
Specific comments
Major concerns
1. ABCG1 antibody specificity
Due to the inconsistency of molecular weight of ABCG1 and occasional multiple bands detected on WB there is a technical concern about the antibody specificity of ABCG1, e.g. SK-N-MC (Figure 4I) two bands with different intensity; TC-32 (Figure 6D) three bands with different intensity; TC-32 (Figure 6E) one band at 100kDa.
Since this aspect can entirely affect the conclusion of this manuscript, this Reviewer suggest to perform additional experiments with western blot to validate the antibody specificity. For example, ABCG1 is transiently knock-downed by siRNA in Ewing sarcoma cells with presumably high ABCG1 protein expression and confirm which bands will be decreased in intensity.
2. Validation of ABCG1 mRNA knockdown by siRNA
The authors need to provide data whether siABCG1 indeed led to decrease of ABCG1 mRNA level.
3. Line numbers: 461-466
If the authors used a non-parametric statistic test such as Mann-Whitney test with two biologically independent experiments, due to the lower statistical power, it is impossible to give a result of P < 0.05.
4 Line numbers: 518-522
The authors report “Using our customised pipeline we identified two FDA approved small molecule inhibitors, allopurinol and phenytoin, that target POU5F1/OCT4 and are used to treat gout and control seizures in epilepsy respectively (Table 2, https://www.dgidb.org/genes/POU5F1#_interactions)“ According to their interaction scores there is neither solid evidence nor strong prediction that allopurinol and phenytoin could inhibit POU5F1/OCT4. The authors need to clearly emphasize and discuss this limitation.
5. References 80 and 81
These refer to a different drug SM-88 not phenytoin.
Minor concerns
1. Line numbers: 435-436
The authors stated “Expression of ABCG1 at the RNA level was confirmed in primary patient-derived ES cells (Figure 6E)”. To this Reviewer the image seems to indicate protein levels.
2. Line numbers: 436- 441
The authors reported “However, there was no correlation between the percentage of CD133-positive cell lines or patient-derived cells and progeny producing ability measured using the colony formation assay in soft agar (R2<0.1) or self-renewing ability from a single cell seeded on low adherent or adherent plates (R2=0.002; Figure 6F), suggesting in patient derived cells CD133 may not identify cell populations with a self-renewing phenotype.” Since this observation can build important evidence that CD133 is not a CSC marker for patient-derived cells, the authors should clearly discuss the possible discrepancy between in vitro cell line models and patient-derived cells.
3. Lines 557-562
The authors state “ABCG1 expression is higher in several cancer types compared to normal tissues [28,55], expression being associated with higher grade tumours [59], metastasis [60] and poor response to chemotherapy [61]. These hypotheses and data are consistent with the premise that ABCG1 identifies ES cells that are capable of surviving chemotherapy and may be responsible for progression and relapse in some patients. We are currently investigating the role of ABCG1 in ES.” Since high ABCG1 expression is associated with better PFS in Ewing sarcoma as demonstrated by the authors, the authors’ argument is not consistent with the observation in other tumor entities, i.e. high ABCG1 expression may render good chemotherapy response in Ewing sarcoma.
Author Response
Please see attachment for a letter to the editor, response to reviewer 1's comments and the edited manuscript file.

Reviewer 2 Report
Excellent manuscript. I will recommend accepting the manuscript.
Author Response
We thank reviewer 2 for their positive comments.